# Physical Exercise Inhibits Inflammation and Microglial Activation

**DOI:** 10.3390/cells8070691

**Published:** 2019-07-09

**Authors:** Onanong Mee-inta, Zi-Wei Zhao, Yu-Min Kuo

**Affiliations:** 1Institute of Basic Medical Sciences, National Cheng Kung University, Tainan 70101, Taiwan; 2Department of Cell Biology and Anatomy, National Cheng Kung University, Tainan 70101, Taiwan

**Keywords:** neuroinflammation, myokine, growth factor, anti-inflammatory, antioxidant, neurodegeneration

## Abstract

Accumulating evidence indicates that exercise can enhance brain function and attenuate neurodegeneration. Besides improving neuroplasticity by altering the synaptic structure and function in various brain regions, exercise also modulates multiple systems that are known to regulate neuroinflammation and glial activation. Activated microglia and several pro-inflammatory cytokines play active roles in the pathogenesis of neurodegenerative diseases, such as Alzheimer’s disease and Parkinson’s disease. The purpose of this review is to highlight the impacts of exercise on microglial activation. Possible mechanisms involved in exercise-modulated microglial activation are also discussed. Undoubtedly, more studies are needed in order to disclose the detailed mechanisms, but this approach offers therapeutic potential for improving the brain health of millions of aging people where pharmacological intervention has failed.

## 1. Introduction

Exercise impacts our body at multiple levels, including the central nervous system (CNS). In responding to exercise-related stress (e.g., hypoxia, heat, free radicals, etc.) and injuries, the body launches multiple endogenous protective and repair systems by altering gene expression and releasing a range of factors that prepare the body for the next challenge. These factors, amongst others, involve trophic effects, anti-oxidation, energy metabolism, and anti-inflammation. Some of these factors enhance brain function and ameliorate brain disorders by inducing neuroplasticity, increasing metabolic efficiency, and improving anti-oxidative capacity [1,2,3,4]. Others maintain brain homeostasis and protect brain from pathological insults by regulating glial activation and neuroinflammation. Activated microglia and several pro-inflammatory cytokines play active roles in the pathogenesis of neurodegenerative diseases, such as Alzheimer’s disease (AD) and Parkinson’s disease (PD). It has been well-documented that although acute, high-intensity exercise may cause muscle injury and induce inflammation, long-term exercise at low-to-moderate intensity negatively regulates the inflammatory response. Here, we review literature that provides empirical evidence showing potential mechanisms for the inflammation-modulating effect of physical exercise, with focus on the effect of anti-microglial activation.

## 2. Microglia Are the Central Modulators of Neuroinflammation

Microglia are the primary immune cells in the CNS. One of their major functions is to maintain the homeostasis of the CNS [5]. In addition to controlling innate and adaptive immune responses in the CNS, microglia also play a significant role in remodeling the dendritic spine formations and modulating neuronal activity [6,7]. In the healthy brain, microglia exist in a resting state characterized by ramified morphology. However, resting microglia are not inactive. Their processes continually survey a defined microenvironment [8,9]. Thus, it has been suggested that “surveying microglia” is a more appropriate name to describe their active surveillance mode [9]. When exposed to pathological insult, microglia transform from a resting stage into a spectrum of activated stages. Activated microglia are characterized by changes in morphology and expression of cytoplasmic and surface proteins [10,11]. Some of the morphological changes (e.g., retraction of ramified processes) and altered protein expressions (e.g., MHC II, CD11b, CD68) have been adopted as markers of microglia activation. The two types of activation states (i.e., the classical activation M1-proinflammatory type and the alternative activation M2-anti-inflammatory type) in macrophages have been introduced into the field of microglia [12,13,14]. Nonetheless, it has been argued whether this M1/M2 polarization occurs in microglia in tissue, especially in the human brains [15].

The activity of microglia can be induced by a plethora of pro-inflammatory factors such as neutrophil-related cytokines, oxidative stress, nucleotide-binding domain, leucine rich containing protein (NLRP-inflammasome), and lipid-mediated proteins derived from metabolic disorders [16,17,18]. Different neurotransmitters are also known to activate microglia via their respective receptors, such as purinergic receptors, glutamate receptors, dopaminergic receptors and cholinergic receptors [9]. The immune cells and molecules (e.g., complement, inflammasome associated protein, gut microbiota, and the bacterial secreted metabolisms [19,20,21]) known to induce peripheral inflammation can also induce microglial activation.

The activity of microglia can be inhibited by anti-inflammatory factors, such as cluster of differentiation-200 (CD200)/CD200 receptor (CD200R), interleukin-10 (IL-10), triggering receptor expressed on myeloid cell-2 (TREM2), and vitamin D3 [22,23,24,25]. These proteins downregulate inflammatory cytokines released by microglia and are involved in the essential signaling pathways in these cells. For example, CD200R can inhibit the activation of the extracellular signal-regulated kinase (ERK), c-Jun N-terminal kinase (JNK), and p38 mitogen-activated protein kinase (MAPK) pathways, which cause cytokine production [26]. IL-10 prevents the lipopolysaccharide (LPS)-induced production and secretion of pro-inflammatory cytokines through the toll-like receptor-4 (TLR4) and nuclear factor kappa B (NF-κB) signaling cascade [27,28]. Moreover, TREM2 downregulates the PI3K/Akt and NF-κB signaling and attenuates the LPS-induced inflammatory responses [29]. Remarkably, expression levels of several anti-microglial activation factors are significantly decreased in AD and PD [30,31]. The deficiency of anti-inflammatory factors results in enhanced microglial activation and subsequently leads to the neuroinflammation, which has been shown to impair in adult hippocampal neurogenesis, neuronal synaptic morphology, and synaptic plasticity [32,33]. Furthermore, it has been hypothesized that inflammation is important for the death of dopaminergic neurons [34]. The levels of the proinflammatory cytokines TNF-α, IFN-γ, IL-1β, and IL-6 are elevated in the cerebrospinal fluid, serum, striatum, and substantia nigra of PD patients [35]. Using positron emission tomography scans with radiotracers for activated microglia and dopamine transporter, a negative correlation between these two markers in the dopaminergic nigrostriatal system has been reported in early PD patients [36], which suggests that microglia are activated early in the disease.

In Table 1, we list studies that have described how different forms of exercise can modulate some of these factors to affect microglia activation.

## 3. Exercise Regulates Microglial Activation by Increasing Anti-Inflammatory Factors

### 3.1. Anti-Inflammatory Cytokines

In response to muscular contractions, myocytes produce and release numerous molecules, termed myokines [56]. Among them, IL-6 was the first identified myokine [56,57,58]. IL-6 has both pro-inflammatory and anti-inflammatory effects. These effects are mediated by distinct signaling pathways: classic-signaling and trans-signaling. The pro-inflammatory effects of IL-6 depend on trans-signaling, in which IL-6 binds to a soluble form of IL-6 receptor; whereas the anti-inflammatory effects of IL-6 depended on classic-signaling, in which IL-6 binds to the membrane-bound non-signaling α-receptor [59,60]. Unlike pro-inflammatory cytokines (e.g., IL-1 and TNFα), IL-6, which can be produced in contracting skeletal muscles and secreted to the vascular system, does not promote major inflammatory mediators [61]. In terms of the anti-inflammatory effects, IL-6 myokine upregulates the expressions of anti-inflammatory cytokine IL-10 and the levels of IL-1 receptor antagonist (IL-1Ra) [39]. Interestingly, the levels of IL-6 are reduced in several neurodegenerative diseases, including AD and PD [60].

In addition to IL-6, other myokines may also have different kinds of anti-inflammatory cytokine properties. It has been shown that long-term exercise can increase the production and secretion of IL-10 in the skeletal muscles [38]. In a peripheral nerve injury mouse model, two weeks of low-intensity exercise was found to inhibit peripheral and central neuroinflammatory responses via upregulation of IL-4 [37]. These anti-inflammatory myokines can be readily transported into the CNS from the peripheral circulation [62]. When IL-10 interacts with its receptor on microglia, it enhances the suppressor of cytokine signaling 3, an inhibitor of cytokine-induced signaling responses resulting in inhibition of inhibits microglial activation [63].

Moreover, exercise can also stimulate the expression of IL-1Ra in the CNS [37]. IL-1Ra has a higher affinity for the IL-1R than IL-1α or IL-1β. Blocking the binding of IL-1 to its receptor interrupts the pro-inflammatory IL-1 signaling cascade and related microglial activity [64]. Therefore, exercise can upregulate the expression of anti-inflammatory cytokines and inhibit microglial activation.

### 3.2. CD200-CD200R

CD200 is an immunomodulatory factor expressed by neurons in the CNS [65]. It inhibits microglial activation through binding with CD200R on microglial plasma membranes [65]. It has been demonstrated that long-term treadmill running increases expression levels of both CD200 and CD200R in mouse brains [40]. When CD200-CD200R interaction occurs, it recruits downstream tyrosine kinases 1 and 2, phosphotyrosine-binding domain proteins. After phosphorylation, these proteins bind to RasGAP and Src homology 2-containing inositol phosphatase, leading to inhibition of downstream Ras/ERK-MAPK inflammatory signaling pathways. Furthermore, CD200/CD200R can also inhibit JNK and p38 MAPK. The inactivation of MAPK signaling results in the suppression of pro-inflammatory cytokine secretion [26,66,67]. Exercise may increase the expression of CD200/CD200R to suppress MAPK signaling pathway and inhibit microglial activation.

### 3.3. TREM2

TREM2, an immunoglobulin superfamily receptor, is mainly expressed by microglia in the CNS [68]. It plays essential roles in microglial phagocytosis and cytoskeleton rearrangement [69]. A soluble form of TREM2, derived from the cell surface receptor, can promote the survival of microglia and their phagocytic activity [70,71]. It has also been suggested that increasing TREM2 levels is an effective therapeutic approach for delaying the pathogenesis of AD [70]. Mutations in TREM2 resulting in loss of function can lead to increased risk of developing AD. Long-term physical exercise increases levels of soluble TREM2 in the CSF of AD patients [41]. When TREM2 forms a complex with DNAX activation proteins 12 (DAP12), a transmembrane adapter protein on the microglial membrane, TREM2 activation causes DAP12 phosphorylation via Src family kinases. The TREM2-DAP12 interaction stimulates PI3K/Akt downstream effect and subsequently blocks MAPK cascade at the RAF level. These signaling pathways lead to the inhibition of inflammatory responses driven by TLR4 in microglia [72,73]. Thus, exercise may regulate the activation status of microglia via upregulation of TREM2.

### 3.4. Heat-Shock Proteins (HSP)

During physical exercise period, neuroprotective HSPs are synthesized and released from skeletal muscle into blood circulation. Hence, HSPs can be considered as myokines. HSPs are categorized into several sub-types (e.g., HSP60, HSP70, and HSP72) [50,51,52,74] and their functions are multifaceted, but mainly in maintaining cellular homeostasis and protein stability [75]. HSPs are associated with endotoxin tolerance response [76]. Noticeably, HSP60 and HSP70 decrease the cellular responses to LPS as well as TLR4 expression, resulting in downregulation of secretion of pro-inflammatory molecules, including TNF-α and IL-1β [77,78,79,80]. Therefore, exercise-induced upregulation of HSPs may negatively regulate the productions of pro-inflammatory cytokines from glial cells.

### 3.5. Metabolic Factors

The efficacy of both acute resistance and chronic endurance exercises enhances the production of Sirtuin-1 (SIRT1), an NAD+-dependent protein deacetylase [44,45,46]. SIRT1 plays critical roles in regulating inflammation and cell survival by deacetylation of transcription factors [81]. Overexpression of SIRT1 deacetylates the NF-κB subunit p65, hence suppresses its activity, resulting in decreased production of proinflammatory cytokines [82,83,84]. Moreover, SIRT1 can activate the expression of transcription factor nuclear factor erythroid 2-related factor 2 (Nrf2) to stimulate the antioxidant response [85]. Nrf2-activated gene expression leads to antioxidant enzymes being released, including glutathione peroxidase and heme oxygenase-1 [86]. Another function of Nrf2 is an antagonist of the NF-κB cascade, which causes reduced production of proinflammatory mediators [87]. More importantly, SIRT1 directly stimulates the transcriptional co-activator peroxisome proliferator-activated receptor gamma coactivator-1-alpha (PGC-1α), which controls inflammation. PGC-1α restrains the activity of NF-κB-mediated microglial activation, but at the same time enhances the phagocytic activity of microglia through the STAT3 and STAT6 pathways [88,89]. Thus, the SIRT1-related Nrf2 and NF-κB pathways may contribute to exercise-induced inhibition of microglial activation and neuroinflammation.

In a diet-induced obesity and leptin resistance mouse model, increased leptin levels are found in the hippocampus [47]. Accumulated leptin induces local astroglial activation and secretion of IL-1β and TNF-α, which subsequently induces microglial activation [47]. Long-term voluntary exercise decreases the degree of microglial activation in the hippocampus by increasing leptin sensitivity and decreasing the levels of IL-1β and TNF-α [47].

### 3.6. Brain-Derived Neurotrophic Factor (BDNF)

BDNF is one of the most studied neurotrophic factors whose production in the brain is increased by exercise [48,49,90]. BDNF is primarily produced by astrocytes and microglia in brain and has multiple effects on neurogenesis and inflammation modulation. Using the 5xFAD AD model mouse which develops severe AD-like pathology at an early age and cognitive dysfunction, it was found that inducing adult hippocampal neurogenesis (AHN) alone did not reverse cognitive dysfunction or pathology, but inducing AHN and BDNF levels through exercise resulted in reduced Aβ load, increased expression of synaptic proteins and improved cognition [90]. These effects were only observed in 5xFAD mice with pathology and not age-matched controls. The exercise induced secretions of BDNF, IL- 6 and fibronectin type III domain–containing protein–5 (FNDC5). It has been demonstrated that muscle contraction-induced activation of PGC-1α stimulates the release of irisin from its membrane-bound precursor FNDC5 [91]. The circulating irisin then enters the brain and upregulates BDNF expression in the brain [91]. Alternatively, ketone body β-hydroxybutyrate released from the liver during exercise has also been shown to induce BDNF expression in the brain [92]. BDNF can regulate brain functions in multiple aspects, including neuronal cell survival, adult hippocampal neurogenesis, and neuroplasticity [93]. Furthermore, BDNF can alleviate microglial activation in several brain disease models [94,95,96,97]. However, the decreased microglial activation is generally considered an indirect consequence of reduced neuronal injury elicited by the BDNF neurotrophic effect, not a direct effect of BDNF.

Nonetheless, we propose that BDNF may directly influence microglial activation through the following pathways. By binding to its receptor TrkB, BDNF can induce ERK activation and CREB phosphorylation, which can block the activity of NF-κB and transcription of certain anti-inflammatory genes [98,99]. Furthermore, BDNF can activate Akt signaling, which suppresses the activity of glycogen synthase kinase 3, resulting in a decrease of NF-κB activation and an increase of CREB activation [100]. BDNF is also known to modulate mitogen-activated protein kinase phosphatase 1, resulting in the reduction of p38 and JNK phosphorylation [101,102,103]. The effect of BDNF on microglial activation requires further in-depth investigation to determine the interactions of the effects of exercise-induced BDNF on neurogenesis and microglia/astrocyte responses.

### 3.7. Antioxidants

It is known that free radical substances are produced during exercise. However, endurance and long-term exercises essentially induce adaptive responses by upregulation of endogenous antioxidants, including glutathione peroxidase (GSH) and superoxide dismutase [53,104,105]. Noticeably, GSH plays vital roles in maintaining redox balance and anti-inflammatory mediators found in astrocytes and microglia [106,107]. The increased GSH synthesis can attenuate the releasing pro-inflammatory factors TNF-α and IL-6. Moreover, it negatively regulates the p38 MAPK, JNK, and NF-κB inflammatory pathways in glial cells [108,109,110], which may also inhibit the production of free radicals. In a 1-methyl-4-phenyl-1,2,3,6-tetrahydropyridine (MPTP)-induced PD mouse study, long-term treadmill running decreased the MPTP-induced upregulation of NADPH oxidase and microglial activation [111]. Therefore, long-term exercise-induced upregulation of antioxidant levels play a crucial role in the protection against neuroinflammation by microglia. The induced upregulation of antioxidants is involved in protection not only to oxidative stress but also to neurodegenerative diseases, including AD and PD [112].

### 3.8. Glymphatic System

The glymphatic system is known as the brain waste clearance system, involving glial cells and lymphatic vessels [113,114]. This system drains the fluid and solutes from the parenchyma (i.e., interstitial fluid) into the cerebrospinal fluid and eventually to the deep cervical lymph nodes [115]. Aquaporin-4 is a key protein component, which is found along the astrocytic endfeet contacting to the cellular membranes of perivascular cells. It functions in brain-water homeostasis and importantly facilitates the pathway of perivascular interstitial fluid-cerebrospinal fluid exchange [116,117]. Impaired glymphatic function is evident in aging and neurodegenerative diseases [118,119]. On the other hand, both voluntary wheel and mandatory treadmill exercise enhanced the efficiency of glymphatic system in normal aging and AD mice [42,120]. Furthermore, the degrees of astrocytic and microglial activations in aged mice were also decreased, whereas the amount of aquaporin-4 increased through exercise [43]. It has been suggested that the anti-microglial activation effect of exercise is due to the enhanced glymphatic system, which facilitates the clearance of metabolic wastes and toxins, such as amyloid-β and α-synuclein, from the brain [121,122].

## 4. Effect of Exercise on Proinflammatory Cytokines and Chemokines

In this section, studies showing that exercise inhibits microglial activation by suppressing the production of pro-inflammatory factors will be discussed. The key studies are summarized in Table 2.

### 4.1. Proinflammatory Cytokines and Chemokines

Exercise can inhibit microglial activation and alleviate pathogenesis of AD and PD in both patients and animal models by downregulating the expression of proinflammatory cytokines [123,124,125,126]. For example, in the APPswe/PS1De9 double-transgenic mouse model of AD, long-term treadmill exercise has been shown to suppress oxidative stress and microglia-induced neuroinflammation by decreasing the level of IL-1β and TNF-α [127]. IL-1β, IL-18, and TNF-α are considered the major proinflammatory cytokines in the CNS [128]. IL-1β and IL-18 can promote inflammasome production to enhance the inflammatory response [116]. In the APP/PS1 model of AD, IL-1β secretion has been shown to enhance NLRP3 inflammasome in microglia, while deficiency of NLRP3 inflammasome decreases deposition of amyloid-β [129]. Furthermore, in several neurological disease models, including multiple sclerosis, PD, and AD, NLRP3 and NLRC4 inflammasomes have been demonstrated to play critical roles in the activation of microglia and astrocytes [18,129,130]. In ovariectomized mice, long-term treadmill running has been shown to decrease the release of IL-1β and IL-18 and inhibit microglial activation from NLRP3 inflammasome production in the hippocampus [123].

Chemokines are responsible for the recruitment of cells of monocytic origin to sites of inflammation, which may aggravate the degrees of local microglial activation [131]. In obese mice induced by a high-fat western diet, expression of a number of proinflammatory molecules, including chemokines CCL2 and CXCL10, are increased in the prefrontal cortex [132]. Long-term wheel running is known to decrease the expression levels of CXCL10 and CCL2, which may lessen the degrees of subsequent microglial activation [132].

### 4.2. Toll-Like Receptor (TLR) Signaling Pathway

TLRs are a class of pattern recognition receptors that play a key role in the innate immune response, including the activation of microglia. Several studies have implied that exercise can inhibit microglial activation by regulating TLR signaling pathways. In the mouse model of PD, soluble α-synuclein indirectly lead to oxidative stress when binding to surface receptors TLR2, TLR4, and CD11b in microglia, resulting in activation of neuroinflammation [133,134,135]. Interestingly, long-term treadmill running suppressed α-synuclein (α-Syn)/ TLR2 mediated neuroinflammation and the associated microglia and NADPH oxidase activation [111]. In another study, long-term treadmill running was found to decrease the high-fat-diet induced hippocampal neuroinflammation, including the degree of microglial activation, levels of proinflammatory cytokines (TNF-α and IL-1β) and cyclooxygenase-2, as well as the expression levels of TLR-4 and its downstream proteins [136]. In these two studies, the authors attributed the exercise-induced anti-microglial activation to downregulation of TLR signaling pathways. However, the actual causal relationship between TLRs and microglial activation remains to be confirmed. 

### 4.3. Gut Microbiota

Bidirectional communication between the brain and the gut has been suggested [138]. The brain can alter gut function by influencing motility, secretion, blood flow, and gut-associated immune function; whereas, the microbiome and products or metabolites secreted from the microbiome can modulate neuronal, immune, metabolic and endocrine pathways [21,139]. It has been suggested that information exchange along the brain-gut-microbe axis can be achieved by various routes, including the vagus nerve, the hypothalamic-pituitary-adrenal axis, and neurotransmitters and hormones, some of which can be produced by the microbiota [140,141]. For example, the gastrointestinal inflammatory tone could cross into brain via vagal afferent neurons [142]. It has been shown that pathogenic gut bacteria, such as *Campylobacter jejuni* or *Salmonella typhimurium* can, via the vagal afferents, induce neuronal activation in the nucleus of the solitary tract, suggesting vagal innervation of the GI tract is capable of detecting host-pathogen interactions in the intestine prior to the challenge producing a systemic response [142].

Intriguingly, gastrointestinal abnormalities such as constipation are highly associated with PD [143]. Recently, studies indicated that α-synuclein accumulation appears early in the gut and propagates via the vagus nerve to the brain in a prion-like fashion [144,145]. In an α-synuclein overexpressed mouse model of PD, α-synuclein pathology and microglial activation are regulated by gut microbiota, whereas antibiotic treatment ameliorates these pathologies [145]. Likewise, the dysregulation of gut microbiota is also known to regulate microglia activation and contribute to AD pathogenesis [146,147,148]. These findings strongly indicate that intestinal dysbiosis can induce microglial activation and other inflammatory responses in the brain.

It has been shown that physical exercise can modify the microbiome and influence the activation status of microglia, as well as performance of learning and memory [137]. However, exactly how exercise can modulate gut microbiota remains unclear, although several potential mechanisms have been suggested. These include indirect effects of exercise on the brain-gut-microbe axis, diet-microbe-host metabolic interactions, neuro-endocrine, and neuro-immune interactions, most likely via the regulation of vagus efferents with respect to the immune system (e.g., spleen and lymph nodes) and gut [142,149,150]

## 5. Conclusions

Undoubtedly, physical exercise can modulate microglial activation in the CNS. Current literature states that low-intensity exercise is sufficient to induce an anti-microglial activation effect by regulating the expressions of various factors. Some of these factors (e.g., myokines) can directly inhibit microglial activation via assorted mechanisms that subsequently prevent neuroinflammation in the CNS. Considerable evidence also suggests that exercise may inhibit microglial activation by downregulation of the levels of pro-inflammatory factors. However, the mechanism for the exercise-related downregulation of pro-inflammatory factors is less clear, as pro-inflammatory cytokines can be secreted from various sources (e.g., injured neurons, astrocytes, and microglia). Thus, the anti-microglial activation effect of exercise can be interpreted indirectly by upregulating the levels of trophic factors, which then lead to reduced neuronal injury and degrees of microglial activation. Furthermore, there are a few reports suggesting that physical exercise can shift the composition of the gut microbiome, which then affects both peripheral and central inflammation, including microglial activation in the CNS. Although some mechanisms are still waiting to be determined, it should be emphasized that physical activity represents a natural strong anti-inflammatory strategy to improve brain function [43].

## Figures and Tables

**Table 1 cells-08-00691-t001:** Exercise regulates microglial activation by increasing anti-inflammatory factors.

	Effects	Models	Time, Frequency& Duration #	Intensity $	Type	Reference
Cytokines	↑ [IL-4]_spinal cord_	Mouse	30 min/day, 5 days/week, Long-term	Low	Treadmill	[37]
	↑ [IL-10]_skeletal muscle_	Rat	60 min/day, 5 days/week, Long-term	Low	Treadmill	[38]
	↑ [IL-6, IL-10]_plasma_	Human	3 h 26 min	N/A	Marathon	[39]
Membrane proteins	↑ [CD200, CD200R]_midbrain_	Mouse	30 min/day, 5 days/week, Long-term	Low & Moderate	Treadmill	[40]
	↑ [TREM2]_CSF_	Human	Long-term	N/A	Physical exercise	[41]
	↑ [Aquaporin-4]_astrocyte_	Mouse	Long-term	N/A	Wheel	[42]
	↑ [Aquaporin-4]_astrocyte_	Rat Mouse	Short- & long-term	High	Treadmill &Wheel	[43]
Metabolism	↑ [SIRT-1]_skeletal muscle_	Rat	60 & 90 min/day, 7 days/week, Long-term	Low & High	Treadmill	[44]
	↑ [SIRT-1]_cardiac muscle_	Rat	20–60 min/day, 5 days/week, Long-term	N/A	Swimming	[45]
	↑ [SIRT-1]_skeletal muscle_	Human	30 s	N/A	Sprint exercise	[46]
	↑ [leptin sensitivity]_hippocampus_	Mouse	Long-term	~5.6 ± 1.2 km/day	Wheel	[47]
Neurotrophic factors	↑ [BDNF]_serum_	Rat	30 min/day, Short-term	Low	Treadmill	[48]
	↑ [BDNF]_serum_	Human	20 min	High	Cycling	[49]
Heat shock proteins	↑ [HSP60]_subcutaneous adipose tissue_	Human	60 min/day, Long-term	Moderate	Aerobic, Treadmill, & Cycling	[50]
	↑ [HSP70]_skeletal muscle_	Rat	60 min/day, Short-term	Mod/High	Treadmill	[51]
	↑ [HSP70]_serum, skeletal muscle_	Human	60 min/day, Short-term	Moderate	Treadmill	[52]
Antioxidant	↑ [Glutathione]_skeletal muscle_	Rat	2 h/day, 5 days/week, Long-term	High	Treadmill	[53]
	↑ [Glutathione]_skeletal muscle_	Dog	40 km/day, 5 days/week, Long-term	High	Treadmill	[53]

# Exercise duration: acute: single bout; short-term: ≤2 weeks; long-term: >2 weeks. $ For human studies, we followed the criteria of Physical Activity Guidelines for Americans [54] to grade the intensity of exercise. For animal studies, we followed the method described in Inoue et al. [55] by using lactate threshold as a criterion to grade the intensity of exercise.

**Table 2 cells-08-00691-t002:** Exercise regulates microglial activation by decreasing pro-inflammatory factors.

	Effects	Models	Time, Frequency & Duration #	Intensity $	Type	Reference
Toll-like receptors	↓ [TLR2]_substantia nigra_	Mouse	60 min/day, 5 days/week, Long-term	Low	Treadmill	[111]
	↓ [TLR4]_hippocampus_	Rat	30 min/day, 5 days/week, Long-term	Low	Treadmill	[136]
Cytokines	↓ [IL-1β, TNF-α]_substantia nigra_	Mouse	60 min/day, 5 days/week, Long-term	Low	Treadmill	[111]
	↓ [IL-1β, TNF-α] _hippocampus_	Mouse	45 min/day, 5 days/week, Long-term	Low	Treadmill	[127]
	↓ [IL-1β, TNF-α]_hippocampus_	Rat	30 min/day, 5 days/week, Long-term	Low	Treadmill	[136]
	↓ [IL-1β, IL-18]_hippocampus_	Mouse	60 min/day, 7 days/week, Long-term	Low	Treadmill	[123]
Chemokines	↓ [CCL2, CXCL10]_prefrontal cortex_	Mouse	30 min/day, Long-term	N/A	Wheel	[132]
Inflammasome	↓ [NLRP3]_hippocampus_	Mouse	60 min/day, 7 days/week, Long-term	Low	Treadmill	[123]
Free radicals	↓ [NADPH oxidase]_substantia nigra_	Mouse	60 min/day, 5 days/week, Long-term	Low	Treadmill	[111]
Microbiome	Shifting gut microbiota	Mouse	60 min/day, 4 days/week, Long-term	Low/High	Treadmill	[137]

# Exercise duration: acute: single bout; short-term: ≤2 weeks; long-term: >2 weeks. $ For human studies, we followed the criteria of Physical Activity Guidelines for Americans [54] to grade the intensity of exercise. For animal studies, we followed the method described in Inoue et al. [55] by using lactate threshold as a criterion to grade the intensity of exercise.

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
