# Peer review of "Physical Exercise Inhibits Inflammation and Microglial Activation"

_cells, 2019, doi:10.3390/cells8070691_

Round 1
Reviewer 1 Report
The manuscript of O. Mee-inta et al. is a very interesting review paper about the interconnection of physical exercise, inflammation and microglial activation. Strong evidence shows that activated microglia and pro-inflammatory cytokines may play crucial roles in the patogenesis of several neurodegenerative diseases. The manuscript reviews the most important literature data on the inflammation-modulating effects of physical exercises and the role of myokines, anti-inflammatory cytokines and other factors on inhibiting microglial activation and provideng neuroprotection.
The reviewer accepts that microglial cells are the central modulators of neuroinflammation. Surveying, ramified microglia play a cental role in pruning of synapses and maintain synaptic plasticity. Under pathological conditions microglia turn to activated ( M1 or M2) stages . Pro-inflammatory factors induce microglia, anti-inflammatory factors inhibit the activity of microglia.
Physical exercises regulate microglial activation by increasing a lot of anti-inflammatory factors (e.g. anti-inflammatory cytokines, CD200, TREM2, Hsp70, SIRT1, BDNF, antioxidants, etc.) and thus inactivate or block several microglial activation pathways. As a consequence, physical exercises are a natural strong anti-infalammatory strategy to prevent and slow down neurodegenerative processes and improve brain function.
The manuscript is well written and summarizes the newest results of the field (65% of the citations are from the last 9 years, between 2011 and 2019). Some typoos occur (e.g.line 99: properties; in reference 1 (line 297) the year 2014 should be bolded, etc.)
I accept the manuscript in present form for publication.
Author Response
Comment #1: The manuscript is well written and summarizes the newest results of the field (65% of the citations are from the last 9 years, between 2011 and 2019). Some typoos occur (e.g.line 99: properties; in reference 1 (line 297) the year 2014 should be bolded, etc.)
Response to Comment #1: Thank you for your comment. We have corrected these typos in the revised manuscript.
Reviewer 2 Report
The review by MEE-INTA et al., discusses the effects of physical exercise upon microglial activation, with a focus on the modulation of inflammatory factors. Some potential mechanisms beyond modulation of pro/anti-inflammatory cytokines through which physical exercise contributes to reducing neuroinflammation and microglial activation (such as metabolic factors, antioxidants, TLRs, glymphatic system and gut microbiota) are also dealt with.
Overall, the manuscript reads well and information is presented concisely. Nonetheless, the relation between all these neuro-inflammation-related factors, physical exercise, and neurodegeneration occurring in PD and AD is poorly approached. In fact, the experimental evidence presented is rather general and not focused on models of AD and PD, which weakens the potential translational significance of physical exercise in neuroinflammation and neurodegeneration. This also affects the novelty and significance of the manuscript when compared to other comprehensive reviews which are available in literature examining the effects of physical exercise on neuro-inflammation occurring in PD and AD (for instance Svensson M, et al. Effects of Physical Exercise on Neuroinflammation, Neuroplasticity, Neurodegeneration, and Behavior: What We Can Learn From Animal Models in Clinical Settings. Neurorehabil Neural Repair. 2015).
A deeper link between neuroinflammation, physical exercise, and neurodegeneration is also important to provide a more comprehensive view of the topic to readers who are not familiar with neuroinflammation. Thus, in the opinion of the reviewer, the manuscript needs to be extensively integrated with additional information linking the mechanisms here described with PD and AD. Several major points listed below should be addressed in order to add consistency to the manuscript.
1) The introduction needs to be better organized avoiding subparagraphs. In the opinion of this reviewer, the information can be fused within a single section where the authors should give some more hints to the role of activated microglia and other pro-inflammatory cytokines in the pathogenesis of neurodegenerative diseases including AD and PD, which seems to be the focus of the review. Likewise, a brief overview of the role of physical exercise in AD and PD should be provided (in either experimental models or patients, if any studies are available).
2) The authors should consistently elaborate and expand with novel references the part stating “Remarkably, expression levels of several anti-microglial activation factors are significantly decreased in AD and PD [30,31]. The deficiency of anti-inflammatory factors results in enhanced microglial activation, which has been shown to impair synaptic plasticity [32]”. Is microglial activation implicated only in impaired synaptic plasticity? This section needs to be extensively integrated with evidence focused on PD and AD.
3) In the single sections dealing with the role of physical activity for each factor here analyzed (microglial activation, cytokines, TLRs, gut microbiota, glymphatic system) the authors should also discuss how these factors are altered specifically in PD and AD, and how physical exercise may yield beneficial effects in PD and AD models. This should be done not only in terms of anti-inflammatory activity but also in neuropathology and behavioral outcomes. Thus, the authors should focus more on PD and AD models both in the main text and in the tables, where the experimental models used to induce neuroinflammation are not reported.
In keeping with this, some recent important studies which are missing need to be included and discussed (see for instance, Real CC et al. Evaluation of exercise-induced modulation of glial activation and dopaminergic damage in a rat model of Parkinson's disease using [(11)C]PBR28 and [(18)F]FDOPA PET. J Cereb Blood Flow Metab. 2019; Song SH et al. Treadmill exercise and wheel exercise improve motor function by suppressing apoptotic neuronal cell death in brain inflammation rats. J Exerc Rehabil. 2018; Real CC et al. Treadmill Exercise Prevents Increase of Neuroinflammation Markers Involved in the Dopaminergic Damage of the 6-OHDA Parkinson's Disease Model. J Mol Neurosci. 2017; Yin M et al. Astroglial water channel aquaporin 4-mediated glymphatic clearance function: A determined factor for time-sensitive treatment of aerobic exercise in patients with Alzheimer's disease. Med Hypotheses. 2018; von Holstein-Rathlou S et al. Voluntary running enhances glymphatic influx in awake behaving, young mice. Neurosci Lett. 2018)
4) The authors should also discuss better the role of gut microbiota concerning its role in neuroinflammation in AD/PD. The mechanisms through which physical activity modulates neuroinflammation through modifying the gut microbiome need to be discussed more in detail.
5) In the opinion of this reviewer, the authors should include in the manuscript a section about the role of aggregated proteins such as beta-amyloid and alpha-synuclein on microglial activation and neuroinflammation in AD and PD. In fact, the spreading of these proteins along the whole gut-brain axis and from neurons to microglia may significantly contribute to neuroinflammation and disease progression (Rocha EM, et al. Alpha-synuclein: Pathology, mitochondrial dysfunction and neuroinflammation in Parkinson's disease. Neurobiol Dis. 2018; Matsumoto J et al. Transmission of α-synuclein-containing erythrocyte-derived extracellular vesicles across the blood-brain barrier via adsorptive mediated transcytosis:another mechanism for initiation and progression of Parkinson's disease? ActaNeuropathol Commun. 2017).
This may be also bound to alterations affecting the cell-clearing systems proteasome and autophagy thus providing a link between altered proteostasis and neuroinflammation (see for instance Limanaqi F, et al. Cell Clearing Systems Bridging Neuro-Immunity and Synaptic Plasticity. Int J Mol Sci. 2019; Limanaqi F, et al. A Sentinel in the Crosstalk Between the Nervous and Immune System: The (Immuno)-Proteasome. Front Immunol. 2019; Savolainen MH, et al. Nigral injection of a proteasomal inhibitor, lactacystin, induces widespread glial cell activation and shows various phenotypes of Parkinson's disease in young and adult mouse. Exp Brain Res. 2017; Plaza-Zabala A et al. Autophagy and Microglia: Novel Partners in Neurodegeneration and Aging. Int J Mol Sci. 2017). On the other hand, physical exercise may foster the clearance of aggregated/toxic proteins thus contributing to alleviating both neuroinflammation and neuropathology (see for instance Jang YC, et al. Association of exercise-induced autophagy upregulation and apoptosis suppression with neuroprotection against pharmacologically induced Parkinson's disease. J Exerc Nutrition Biochem. 2018.; Hwang DJ, et al. Neuroprotective effect of treadmill exercise possibly via regulation of lysosomal degradation molecules in mice with pharmacologically induced Parkinson's disease. J Physiol Sci. 2018 Sep;68(5):707-716).
Author Response
Comment #1: The introduction needs to be better organized avoiding subparagraphs. In the opinion of this reviewer, the information can be fused within a single section where the authors should give some more hints to the role of activated microglia and other pro-inflammatory cytokines in the pathogenesis of neurodegenerative diseases including AD and PD, which seems to be the focus of the review. Likewise, a brief overview of the role of physical exercise in AD and PD should be provided (in either experimental models or patients, if any studies are available).
Response to Comment #1: This review focuses on the beneficial effects of exercise on microglial activation. Because microglial activation plays important roles in the pathogenesis of AD and PD, we used these two diseases as examples to address the effects of exercise. The roles of microglia in the pathogenesis of AD and PD have been extensively discussed elsewhere and are not the main objective of this review. In response to Reviewer 2’s comment, we have modified the subheading and increased materials regarding the effect of physical exercise on AD and PD throughout the manuscript.
Comment #2. The authors should consistently elaborate and expand with novel references the part stating “Remarkably, expression levels of several anti-microglial activation factors are significantly decreased in AD and PD [30,31]. The deficiency of anti-inflammatory factors results in enhanced microglial activation, which has been shown to impair synaptic plasticity [32]”. Is microglial activation implicated only in impaired synaptic plasticity? This section needs to be extensively integrated with evidence focused on PD and AD.
Response to Comment #2: As suggested by the Reviewer 2, we have expanded the part as shown below:
Line 73-83: Remarkably, expression levels of several anti-microglial activation factors are significantly decreased in AD and PD [30,31]. The deficiency of anti-inflammatory factors results in enhanced microglial activation and subsequently leads to the neuroinflammation, which has been shown to impair in adult hippocampal neurogenesis, neuronal synaptic morphology, and synaptic plasticity [32,33]. Furthermore, it has been hypothesized that inflammation is important for the death of dopaminergic neurons [34]. The levels of the proinflammatory cytokines TNF-α, IFN-γ, IL-1β, and IL-6 are elevated in the cerebrospinal fluid, serum, striatum, and substantia nigra of PD patients [35]. Using positron emission tomography scans with radiotracers for activated microglia and dopamine transporter, a negative correlation between these two markers in the dopaminergic nigrostriatal system has been reported in early PD patients [36], which suggests that microglia are activated early in the disease.
Comment 3. In the single sections dealing with the role of physical activity for each factor here analyzed (microglial activation, cytokines, TLRs, gut microbiota, glymphatic system) the authors should also discuss how these factors are altered specifically in PD and AD, and how physical exercise may yield beneficial effects in PD and AD models. This should be done not only in terms of anti-inflammatory activity but also in neuropathology and behavioral outcomes. Thus, the authors should focus more on PD and AD models both in the main text and in the tables, where the experimental models used to induce neuroinflammation are not reported.
In keeping with this, some recent important studies which are missing need to be included and discussed (see for instance, Real CC et al. Evaluation of exercise-induced modulation of glial activation and dopaminergic damage in a rat model of Parkinson's disease using [(11)C]PBR28 and [(18)F]FDOPA PET. J Cereb Blood Flow Metab. 2019; Song SH et al. Treadmill exercise and wheel exercise improve motor function by suppressing apoptotic neuronal cell death in brain inflammation rats. J Exerc Rehabil. 2018; Real CC et al. Treadmill Exercise Prevents Increase of Neuroinflammation Markers Involved in the Dopaminergic Damage of the 6-OHDA Parkinson's Disease Model. J Mol Neurosci. 2017; Yin M et al. Astroglial water channel aquaporin 4-mediated glymphatic clearance function: A determined factor for time-sensitive treatment of aerobic exercise in patients with Alzheimer's disease. Med Hypotheses. 2018; von Holstein-Rathlou S et al. Voluntary running enhances glymphatic influx in awake behaving, young mice. Neurosci Lett. 2018)
Response to Comment #3: As aforementioned, the roles of microglia in the pathogenesis of AD and PD have been extensively discussed elsewhere and are not the main objective of this review. Nonetheless, we still expanded the parts to link functional roles of neuroinflammation/microglial activation in AD and PD with the suggested references:
Line 100-101: Interestingly, the levels of IL-6 are reduced in several neurodegenerative diseases, including AD and PD [60].
Line 220-226: On the other hand, both voluntary wheel and mandatory treadmill exercise enhanced the efficiency of glymphatic system in normal aging and AD mice [42,120]. Furthermore, the degrees of astrocytic and microglial activations in aged mice are also decreased, whereas the amount of aquaporin-4 is increased by exercise [43]. It has been suggested that the anti-microglial activation effect of exercise is due to the enhanced glymphatic system, which facilitate the clearance of metabolic wastes and toxins, such as amyloid-β and α-synuclein, from the brain [121,122].
Line 232-236: Exercise can inhibit microglial activation and alleviate pathogenesis of AD and PD in both patients and animal models by downregulating the expression of proinflammatory cytokines [123-126]. For example, in the APPswe/PS1De9 double-transgenic mouse model of AD, long-term treadmill exercise has been shown to suppress oxidative stress and microglia-induced neuroinflammation via decreasing the level of IL-1β and TNF-α [127].
Line 254-259: Several studies have implied that exercise can inhibit microglial activation by regulating TLR signaling pathways. In the mouse model of PD, soluble α-synuclein indirectly lead to oxidative stress when binding to surface receptors TLR2, TLR4, and CD11b in microglia, resulting in activation of neuroinflammation [133-135]. Interestingly, long-term treadmill running suppressed α-synuclein (α-Syn-)/ TLR2 mediated neuroinflammation and the associated microglia and NADPH oxidase activation [111].
Line 269-295: Bidirectional communication between the brain and the gut has been suggested [138]. The brain can alter gut function by influencing motility, secretion, blood flow, and gut-associated immune function; whereas, the microbiome and products or metabolites secreted from the microbiome can modulate neuronal, immune, metabolic and endocrine pathways [21,139]. It has been suggested that information exchange along the brain-gut-microbe axis can be achieved by various routes, including the vagus nerve, the hypothalamic-pituitary-adrenal axis, and a sort of neurotransmitters and hormones, some of which can be produced by the microbiota [140,141]. For example, the gastrointestinal inflammatory tone could cross into brain via vagal afferent neurons [142]. It has been shown that pathogenic gut bacteria, such as Campylobacter jejuni or Salmonella typhimurium can, via the vagal afferents, induce neuronal activation in the nucleus of the solitary tract, suggesting vagal innervation of the GI tract is capable of detecting host-pathogen interactions in the intestine prior to the challenge producing a systemic response [142].
Intriguingly, gastrointestinal abnormalities such as constipation is highly associated with PD [143]. Recently studies indicated that α-synuclein accumulation appear early in the gut and propagates via the vagus nerve to the brain in a prion-like fashion [144,145]. In an α-synuclein overexpressed mouse model of PD, α-synuclein pathology and microglial activation are regulated by gut microbiota, whereas antibiotic treatment ameliorates these pathologies [145]. Likewise, the dysregulation of gut microbiota are also known to regulate microglia activation and contribute to AD pathogenesis [146-148]. These findings strongly indicate that intestinal dysbiosis can induce microglial activation and other inflammatory responses in the brain.
It has been shown that physical exercise can modify the microbiome and influence the activation status of microglia, as well as performance of learning and memory [137]. However, exactly how exercise can modulate gut microbiota remains unclear, although several potential mechanisms have been suggested. These include indirect effects of exercise on the brain-gut-microbe axis, diet-microbe-host metabolic interactions, neuro-endocrine and neuro-immune interactions, most likely via the regulation of vagus efferents to the immune system (e.g., spleen and lymph nodes) and to the gut [142,149,150].
Comment 4) The authors should also discuss better the role of gut microbiota concerning its role in neuroinflammation in AD/PD. The mechanisms through which physical activity modulates neuroinflammation through modifying the gut microbiome need to be discussed more in detail.
Response to Comment #4: As suggested, we have expanded this section in the revised manuscript with emphasis on PD and AD.
Comment 5) In the opinion of this reviewer, the authors should include in the manuscript a section about the role of aggregated proteins such as beta-amyloid and alpha-synuclein on microglial activation and neuroinflammation in AD and PD. In fact, the spreading of these proteins along the whole gut-brain axis and from neurons to microglia may significantly contribute to neuroinflammation and disease progression (Rocha EM, et al. Alpha-synuclein: Pathology, mitochondrial dysfunction and neuroinflammation in Parkinson's disease. Neurobiol Dis. 2018; Matsumoto J et al. Transmission of α-synuclein-containing erythrocyte-derived extracellular vesicles across the blood-brain barrier via adsorptive mediated transcytosis:another mechanism for initiation and progression of Parkinson's disease? ActaNeuropathol Commun. 2017).
This may be also bound to alterations affecting the cell-clearing systems proteasome and autophagy thus providing a link between altered proteostasis and neuroinflammation (see for instance Limanaqi F, et al. Cell Clearing Systems Bridging Neuro-Immunity and Synaptic Plasticity. Int J Mol Sci. 2019; Limanaqi F, et al. A Sentinel in the Crosstalk Between the Nervous and Immune System: The (Immuno)-Proteasome. Front Immunol. 2019; Savolainen MH, et al. Nigral injection of a proteasomal inhibitor, lactacystin, induces widespread glial cell activation and shows various phenotypes of Parkinson's disease in young and adult mouse. Exp Brain Res. 2017; Plaza-Zabala A et al. Autophagy and Microglia: Novel Partners in Neurodegeneration and Aging. Int J Mol Sci. 2017). On the other hand, physical exercise may foster the clearance of aggregated/toxic proteins thus contributing to alleviating both neuroinflammation and neuropathology (see for instance Jang YC, et al. Association of exercise-induced autophagy upregulation and apoptosis suppression with neuroprotection against pharmacologically induced Parkinson's disease. J Exerc Nutrition Biochem. 2018.; Hwang DJ, et al. Neuroprotective effect of treadmill exercise possibly via regulation of lysosomal degradation molecules in mice with pharmacologically induced Parkinson's disease. J Physiol Sci. 2018 Sep;68(5):707-716).
Response to Comment #5: As aforementioned, the roles of microglia in the pathogenesis of AD and PD have been extensively discussed elsewhere and are not the main objective of this review. Nonetheless, we still made changes to highlight the functional roles of neuroinflammation/microglial activation in AD and PD. Please see the responses to Comment #3.
Reviewer 3 Report
Review paper: "Physical exercise of inhibition inflammation and microglial activation" is a very interesting work that raises the topic that is important for a wide range of scientists. The work is well-written, transparent and extremely important for studying the mechanism of the potential impact of exercise on neurodegenerative diseases.
I only recommend to expand section 3.3 "Gut microbiota". In my opinion there is too little information about the impact of the exercise on gut microbiota and the role of microbiota on microglia. I recommend the following publications, which may be helpful: PMID: 286 40 632 and 25800089.
Generally, I'm glad that I could review this work.
Author Response
Reviewer 3
Comment #1. I only recommend to expand section 3.3 "Gut microbiota". In my opinion there is too little information about the impact of the exercise on gut microbiota and the role of microbiota on microglia. I recommend the following publications, which may be helpful: PMID: 286 40 632 and 25800089.
Response to Comment #1: As suggested, we have expanded this section in the revised manuscript:
Line 269-295: Bidirectional communication between the brain and the gut has been suggested [138]. The brain can alter gut function by influencing motility, secretion, blood flow, and gut-associated immune function; whereas, the microbiome and products or metabolites secreted from the microbiome can modulate neuronal, immune, metabolic and endocrine pathways [21,139]. It has been suggested that information exchange along the brain-gut-microbe axis can be achieved by various routes, including the vagus nerve, the hypothalamic-pituitary-adrenal axis, and a sort of neurotransmitters and hormones, some of which can be produced by the microbiota [140,141]. For example, the gastrointestinal inflammatory tone could cross into brain via vagal afferent neurons [142]. It has been shown that pathogenic gut bacteria, such as Campylobacter jejuni or Salmonella typhimurium can, via the vagal afferents, induce neuronal activation in the nucleus of the solitary tract, suggesting vagal innervation of the GI tract is capable of detecting host-pathogen interactions in the intestine prior to the challenge producing a systemic response [142].
Intriguingly, gastrointestinal abnormalities such as constipation is highly associated with PD [143]. Recently studies indicated that α-synuclein accumulation appear early in the gut and propagates via the vagus nerve to the brain in a prion-like fashion [144,145]. In an α-synuclein overexpressed mouse model of PD, α-synuclein pathology and microglial activation are regulated by gut microbiota, whereas antibiotic treatment ameliorates these pathologies [145]. Likewise, the dysregulation of gut microbiota are also known to regulate microglia activation and contribute to AD pathogenesis [146-148]. These findings strongly indicate that intestinal dysbiosis can induce microglial activation and other inflammatory responses in the brain.
It has been shown that physical exercise can modify the microbiome and influence the activation status of microglia, as well as performance of learning and memory [137]. However, exactly how exercise can modulate gut microbiota remains unclear, although several potential mechanisms have been suggested. These include indirect effects of exercise on the brain-gut-microbe axis, diet-microbe-host metabolic interactions, neuro-endocrine and neuro-immune interactions, most likely via the regulation of vagus efferents to the immune system (e.g., spleen and lymph nodes) and to the gut [142,149,150].
Round 2
Reviewer 2 Report
The authors addressed most of the comments raised by this reviewer.